# Polysialic Acid Sustains the Hypoxia-Induced Migration and Undifferentiated State of Human Glioblastoma Cells

**DOI:** 10.3390/ijms23179563

**Published:** 2022-08-24

**Authors:** Paolo Rosa, Sofia Scibetta, Giuseppe Pepe, Giorgio Mangino, Luca Capocci, Sam J. Moons, Thomas J. Boltje, Francesco Fazi, Vincenzo Petrozza, Alba Di Pardo, Vittorio Maglione, Antonella Calogero

**Affiliations:** 1Department of Medical-Surgical Sciences and Biotechnologies, University of Rome “Sapienza”, Polo Pontino, C.so della Repubblica 79, 04100 Latina, Italy; 2IRCCS Neuromed, Via Dell’Elettronica, 86077 Pozzilli, Italy; 3Institute for Molecules and Materials, Radboud University, Heyendaalseweg 135, 6525 AJ Nijmegen, The Netherlands; 4Department of Anatomical, Histological, Forensic & Orthopedic Sciences, Section of Histology & Medical Embryology, University of Rome “Sapienza”, Via A. Scarpa, 14-16, 00161 Rome, Italy; 5ICOT, Istituto Chirurgico Ortopedico Traumatologico, Via F. Faggiana 1668, 04100 Latina, Italy

**Keywords:** glioblastoma, hypoxia, polysialic acid, migration, differentiation

## Abstract

Gliomas are the most common primary malignant brain tumors. Glioblastoma, IDH-wildtype (GBM, CNS WHO grade 4) is the most aggressive form of glioma and is characterized by extensive hypoxic areas that strongly correlate with tumor malignancy. Hypoxia promotes several processes, including stemness, migration, invasion, angiogenesis, and radio- and chemoresistance, that have direct impacts on treatment failure. Thus, there is still an increasing need to identify novel targets to limit GBM relapse. Polysialic acid (PSA) is a carbohydrate composed of a linear polymer of α2,8-linked sialic acids, primarily attached to the Neural Cell Adhesion Molecule (NCAM). It is considered an oncodevelopmental antigen that is re-expressed in various tumors. High levels of PSA-NCAM are associated with high-grade and poorly differentiated tumors. Here, we investigated the effect of PSA inhibition in GBM cells under low oxygen concentrations. Our main results highlight the way in which hypoxia stimulates polysialylation in U87-MG cells and in a GBM primary culture. By lowering PSA levels with the sialic acid analog, F-NANA, we also inhibited GBM cell migration and interfered with their differentiation influenced by the hypoxic microenvironment. Our findings suggest that PSA may represent a possible molecular target for the development of alternative pharmacological strategies to manage a devastating tumor like GBM.

## 1. Introduction

Gliomas account for the great majority of primary tumors (30–50%) that arise within the central nervous system (CNS) in adult age [1]. Glioblastoma, IDH-wildtype [2] (GBM) is the most common and aggressive form of glioma (CNS WHO grade 4) with a median survival of approximately 20 months in patients who undergo total surgical resection, radio- and chemotherapy [3]. It is often resistant to treatment. The standard treatment option is the use of Temozolomide (TMZ), which is capable of slowing GBM cell growth by inducing apoptosis [4], autophagy [5], or by interfering with the cell cycle [6].

The failure of current therapies is mainly to be ascribed to the extraordinary capacity of GBM cells to migrate and invade the surrounding brain parenchyma as well as resistance to standard treatments. Like other solid tumors, GBM is characterized by extensive hypoxic areas that strongly correlate with tumor grade. Low oxygen levels have indeed been demonstrated to promote radio- and chemoresistance, invasiveness and angiogenesis [7]. Further, hypoxia is able to favor stemness by forcing GBM cells to dedifferentiate [8,9]. In recent years, many efforts have been made to find potential therapies based on cell differentiation [10,11].

Polysialic acid (PSA) is defined as a unique carbohydrate of a linear homopolymer of α2,8-linked sialic acid. In neural cells, PSA is found primarily attached to N-glycans of the Neural Cell Adhesion Molecule (NCAM, also known as CD56) and is particularly abundant in the embryonic and early postnatal brain [12]. In the adult brain, PSA is mainly observed in those brain regions where neurogenesis persists, namely sub-ventricular zone (SVZ), sub-granular zone (SGZ) and olfactory bulbs (OB) [13]. The polysialylated form of NCAM plays an important role in neural development, being involved in cell migration [14], synaptic plasticity [15,16] and neurite outgrowth [17]. NCAM undergoes post-translational modifications being polysialylated by two polysialyltransferases (PolySTs) in the Golgi, STX (ST8SiaII), and PST (ST8SiaIV) [18]. PSA-NCAM is highly expressed during development, and is re-expressed in many cancers. Due to its anti-adhesive and repulsive characteristics as well as to its ability to modulate signaling, PSA plays a pivotal role in cancer cell detachment and metastasis [19]. Indeed, high content of PSA-NCAM is associated with high-grade tumors characterized by undifferentiated cells which spread aggressively. Many strategies have been explored to lower PSA expression either in vitro or in vivo. Silencing PolySTs or overexpressing enzymes that degrade sialic acids efficiently reduce the degree of polysialylation of NCAM with a benefit against cancer cell motility and metastasis [20]. Specific novel inhibitors have been developed to selectively target sialic acids biosynthesis. Among others, F-NANA (Ac_5_3F_ax_Neu5Ac) has shown promising potential in its capability to efficiently block sialylation in many cancer types [21,22,23,24,25]. Although the overexpression of PSA has been associated with high-grade tumors, its role in GBM pathogenesis as well as its behavior in a hypoxic microenvironment is still poorly understood and needs further investigations

In light of these considerations, in this study we investigated the role of PSA in GBM cells exposed to low oxygen concentrations by analyzing the effect of the inhibition of its levels in the resistance to TMZ, migration and differentiation.

We also demonstrated that the administration of F-NANA in vivo is possible and is associated with a significant reduction of PSA in the brain of wild type mice, providing the first evidence that the compound may be infused in vivo and may potentially represent a possible therapeutic agent for the treatment of GBM.

## 2. Results

### 2.1. PSA Expression in GBM Cell Cultures and Hypoxic Areas of GBM Tissues

Polysialylation of NCAM has been associated with many types of cancers, especially high degree malignancies [26,27]. Given the small number of studies concerning the role of PSA in GBM, we investigated its expression in the cell lines mainly used in GBM research. As shown in Figure 1A levels of PSA were particularly high in the human GBM cell lines, U251 and U373, compared to U87-MG, A172 and the mouse glioma cell line GL261. We have also analyzed NCAM expression, which was present in the above-mentioned cell lines in different isoforms, with different expression levels. For example, U87-MG cells showed only the presence of the 140 and 180 kDa isoforms, the expression of which is low. To note, the abundance of PSA-NCAM expressions in the neonatal rat astrocytes homogenate, which was included as positive control. Further, we have analyzed the expression of the two polysialyltransferase genes responsible of the polymerization of sialic acid monomers in GBM primary cultures. The levels of ST8SiaII and IV mRNAs were variably expressed among the different GBM patients, compared to low grade glioma primary cultures, to a metastasis and to the GBM cell lines, U87-MG and U251 (Figure 1B). Finally, the presence of PSA and NCAM proteins were investigated in hypoxic areas of GBM, where HIF-1α protein showed a marked nuclear signal. In those areas, PSA showed a particularly strong immunoreactivity in the membrane and cytoplasm of cells in the GBM tissue, compared to a normal, non-tumoral brain section. Similarly, NCAM expression results were both cytoplasmatic and membranous, but with a weaker signal (Figure 1C).

### 2.2. PSA Levels Are Induced by Chronic Hypoxia in U87-MG Cells

The tumor microenvironment of a solid tumor as GBM is characterized by extensive hypoxia, which is a condition that favors stemness and migration of cancer cells [28,29,30]. Since PSA is involved in cell differentiation and motility [31,32], we have investigated its levels during chronic hypoxia in GBM cells. As shown in Figure 2A the induction of ST8SiaII and IV gene expressions occurred after 48 h of hypoxia, and were about 1.5 and 2-fold higher, respectively, compared to U87-MG cells grown in normoxia. Then, the chronic hypoxia conditions of our model were confirmed by the strong induction of HIF-1α, by Western blot analysis, after 6 h (Figure 2B). Next, we investigated the expression of total NCAM and PSA in U87-MG cells grown under low oxygen concentrations. Our results reveal the induction of polysialylation after 48 and 72 h (about 1.4 and 1.6-fold higher, respectively) of the NCAM protein, with no changes in the levels. Finally, we investigated the expression of extracellular NCAM and its polysialylated form in U87-MG cells under hypoxia. As reported in Figure 2C-D plasma membrane localization of NCAM is not influenced by chronic hypoxia (59.6 ± 1.1% in normoxia vs. 57.8 ± 2.3% under hypoxia). Conversely, PSA-NCAM was expressed on the membrane of a greater number of U87-MG cells grown under normal oxygen concentrations compared to cells grown for 72 h under low oxygen levels (51.8 ± 1.7% in normoxia vs. 37.5 ± 2.1% under hypoxia, *p* < 0.05).

### 2.3. F-NANA Reduces Hypoxia-Induced PSA Levels with No Effect on Proliferation and Chemoresistance to TMZ

In order to study the role of PSA in GBM cells under hypoxia we treated U87-MG cells with F-NANA, a cell permeable analog of sialic acid, able to competitively inhibit sialyltransferase activity including polysialyltransferases [20,21,22]. As reported in Figure 3A, treatment of U87-MG cells with 100 μM F-NANA for 72 h totally inhibited PSA levels in conditions of normal oxygen concentrations. Moreover, the same treatment was efficient to reduce the hypoxia-induced expression of PSA to levels below control U87-MG cells. No changes in NCAM expression were observed. Further, we studied the effect of PSA inhibition on proliferation and chemoresistance of U87-MG cells under hypoxia. Treatment with F-NANA did not show any effect on slowing the proliferation of U87-MG cells in both normoxic and hypoxic conditions (Figure 3B). In the same way, Figure 3C reports that treatment with F-NANA was not able to sensitize U87-MG cells and to overcome hypoxia-induced resistance to chemotherapy with TMZ.

### 2.4. F-NANA Inhibits Hypoxia-Induced Migration of U87-MG Cells and Is Effective in a GBM Primary Culture

Several studies have demonstrated the role of the polysialylation of NCAM on cell migration [12,33]. Here, we investigated the role of PSA in the migration of human GBM cells under low oxygen concentrations. As reported in Figure 4A, inhibition of PSA by F-NANA was able to reduce the migratory ability of U87-MG cells stimulated by low oxygen concentrations of about 40%. Then, as shown in Figure 4B, we confirmed the effect of hypoxia in inducing PSA levels in a self-established GBM primary culture (GL18-15), with levels 2-fold higher compared to normoxic cells. Further, the treatment of GL18-15 primary cells with F-NANA strongly reduced PSA levels both under normoxia and hypoxia, respectively of about 80 and 90%. The reduction of PSA levels can also be appreciated from NCAM expression analysis, concomitant with the increase of the lower molecular weight 140 kDa isoform. Interestingly, the reduction of PSA levels, upon F-NANA administration, was associated with a significant inhibition of the migration ability of GL18-15 cells under hypoxia (Figure 4C).

### 2.5. PSA Reduction Interferes with the Hypoxia-Induced De-Differentiation of U87-MG Cells

A number of studies have highlighted the role of PSA in cellular differentiation [16]. The hypoxic microenvironment within the cells of a solid tumor such as GBM, can push these cells to de-differentiate [9]. Figure 5A reports the time-course of U87-MG cells grown for 72 h under hypoxia and highlights the induction of the stemness genes Oct-4, Sox-2 and Nanog. While the expression for Oct-4 and Sox-2 was highest after 12 h (about a 1.8 and 1.4-fold increase, respectively), the maximal expression for Nanog was reached after 24 h of hypoxia, with a more than 2-fold induction compared to control cells grown in conditions of normal oxygen concentrations.

Treatment with F-NANA significantly prevented the induction of the stemness genes Oct-4 and Nanog (Figure 5B). The ability of F-NANA to interfere with the hypoxia-induced de-differentiation of U87-MG cells was confirmed by analyzing protein expression of the stemness marker nestin and the differentiation markers GFAP and βIII-tubulin after 72 h of low oxygen tensions. As reported in Figure 5C, hypoxic conditions were associated with increased levels of nestin and with a reduction of βIII-tubulin expression. Treatment with F-NANA, by lowering PSA content, induced a significant reduction of nestin and a small increase of βIII-tubulin protein levels in both normoxic and hypoxic conditions. The analysis on GFAP protein expression did not show any significative changes. Finally, the inverse correlation between PSA and βIII-tubulin was confirmed by immunofluorescence analysis (Figure 5D).

### 2.6. PSA Levels Decrease upon GBM Stem-like Cell Differentiation

Among the many causes of GBM resistance and tumor relapse, cancer stem cells play a pivotal role. Undifferentiated glioma stem cells (GSCs) are localized in the tumor within areas not easily accessible to nutrients, oxygen and, inevitably, anticancer treatments [34,35,36]. In light of these considerations, we investigated any variation in PSA metabolism in the differentiation of U87-derived neurospheres (U87NS), a cellular model of GBM stem-like cells. To this regard, U87NS cells were differentiated for 5 days by removing the stem cell-permissive medium and adding the medium supplemented with 10% FBS (Figure 6A). During this differentiation assay the switch from the floating growth modality to the acquisition of the attachment to the substrate capability is usually observed. Thus, we investigated the expression of the PolySTs during the differentiation of U87NS. As reported in Figure 6B, gene expression levels of both ST8SiaII and ST8SiaIV decreased during cell differentiation, in U87NS, this was associated with a significant reduction of PSA levels, as assessed by immunoblotting (Figure 6C). Finally, F-NANA treatment is efficient in reducing PSA levels during differentiation (Figure 6D). This reduction was also appreciable by the lowering of the levels of the 180 kDa isoform of NCAM and the increase of its 140 kDa isoform.

### 2.7. PSA Sustains the Undifferentiated State of GBM Stem-like Cells

The abundance of the polysialylated form of NCAM has been well documented for less differentiated cells [32]. In order to confirm a role for PSA in the maintenance of an undifferentiated state of GBM stem-like cells, we treated U87NS cells with F-NANA upon differentiation. As shown in Figure 7A, inhibition of PSA was associated with the downregulation of the expression of the stemness genes Oct-4 and Nanog. No effect was observed for Sox-2. Moreover, the immunofluorescence analysis shown in Figure 7B–D highlighted the decrease of PSA and nestin signals concomitant to the increase of GFAP levels upon differentiation of U87NS cells. Treatment with F-NANA caused a greater increase in the number of cells positive for GFAP and negative for nestin, thus favoring U87NS cell differentiation. This process seemed not to be involving the expression of βIII-tubulin (Appendix A).

### 2.8. Intranasal Administration of F-NANA Reduces PSA Levels in the Brain of Wild Type Mice

In order to provide a proof of concept that F-NANA administration might be feasible in vivo, we treated four-week-old B6CBAF1/J wild type (WT) mice, for three weeks with 100 μM F-NANA. Immunoblotting analysis of mouse brain tissues showed that treatment with the compound significantly reduced levels of PSA (Figure 8A), with no effect on the expression of NCAM (Figure 8B). Interestingly, F-NANA did not exert any detectable effect on the global sialylation pattern of other brain proteins, as assessed by SNA and MALII immunoblottings (Figure 8C,D). Importantly, no effects were observed in terms of body weight and motor performance, assessed by Rotarod test (Figure 8E,F), suggesting that the treatment did not exert any detrimental effect on mouse well-being.

## 3. Discussion

The main findings reported in this study show that: (i) the hypoxic microenvironment stimulates the production of PSA in GBM cells; (ii) PSA plays a role in the migration of GBM cells under hypoxia; (iii) PSA is involved in the hypoxia-induced de-differentiation of GBM cells; (iv) GBM-stem like cells rely on PSA expression to maintain an undifferentiated state; (v) the modulation of PSA levels is possible in vivo.

The role of the polysialylated form of NCAM has already been demonstrated to be associated to embryonic brain development [37], cellular differentiation [31], migration [38] and metastasis of cancer cells [39]. However, the involvement of PSA in an aggressive tumor such as GBM, which takes advantage of low oxygen concentrations to resist therapies, move, invade the surrounding brain parenchyma, and remain in an undifferentiated state, has been poorly investigated.

Given the well characterized roles of the polysialylation of NCAM in either migration and differentiation, our aim was to investigate its involvement in GBM cell behavior under hypoxia, a condition that favors these processes and that we hypothesize could have an influence on PSA levels to sustain tumor aggressiveness. To our knowledge this is the first study exploring the effects of the hypoxic microenvironment on the polysialylation of NCAM in human GBM cells.

Firstly, we qualitatively show the presence of the isoforms of NCAM and the relative degree of polysialylation in the most commonly used and cited GBM cell lines. The variability that we report could be associated with either the different genetic background of the cells or their different degree of differentiation (see also Appendix A). Also, the mRNA expression analysis of the PolySTs performed on patient-derived GBM primary cultures confirmed this variability. In glioma, ST8SiaIV was demonstrated to be the most responsible for PSA biosynthesis, thus conferring to cells invasive characteristics [33]. Further, ST8SiaII was demonstrated to synthesize shorter polymers of PSA on NCAM compared to ST8SiaIV, and can thus be considered a promising target for metastatic tumors [12]. Since PSA levels have been associated with high-grade tumors, it would be important to study their dynamics, as well as the balance between the PolySTs in different GBM patients, trying to understand their relevance among tumor grades and whether their levels can help monitoring and predicting the response to therapies. Furthermore, we found PSA-NCAM expressed in those areas of GBM tissues positive for HIF-1α, underlying our hypothesis that the hypoxic microenvironment could stimulate PSA expression and have a physiopathological relevance in vivo.

Therefore, in the present work we demonstrated that hypoxia induces the levels of PSA in U87-MG cells, which represented the ideal model for this study, being characterized by low expression of NCAM and its polysialylated form. Hypoxia, also, induces the levels of PolySTs II and IV, enzymes responsible for the synthesis and elongation of long chains of sialic acids [18]. In particular, we observe the rise in ST8SiaII and IV mRNAs expression as well as in PSA levels without changes in NCAM expression, highlighting the way in which low oxygen concentrations are responsible of an augmented degree of NCAM polysialylation in these cells. However, studies on the role of PSA in cancer cells under low oxygen concentrations are still lacking. The only study to mention is the work by Elkashef and colleagues [40], who demonstrated that PSA sustains cell survival and migration of neuroendocrine tumor cells under hypoxia. Nevertheless, their investigations were conducted in the neuroblastoma cell line SH-SY5Y and the rat glioma cell line C6, which was forced to express ST8SiaII for PSA biosynthesis (C6-STX). Here, we observed that U87-MG cells under hypoxic conditions upregulate the total levels of PSA, which are retained inside the cell (see also Appendix A). The expression of the polysialylated form of NCAM on the membrane of cells has various roles. PSA exerts repulsive forces on trans-homophilic NCAM–NCAM interactions leading to mechanisms such as cell adhesion [19] and cell migration [14]. Further, PSA-NCAM on tumor cells may help them escape immunosurveillance (likely PSA expressed on bacteria), thus favoring chemoresistance [41]. We already know that PSA on the extracellular side of cancer cells favors metastases [39], and that hypoxia stimulates the migration of GBM cells [8,42]. One possible explanation for this apparent contradictory situation is that hypoxic microenvironments cause a change of the extracellular pH, which in turn alters extracellular charges [43,44]. It therefore becomes plausible to speculate that in conditions of low oxygen concentrations PSA expression needs to be elevated and confined inside the cell for an energetic and/or electrostatically reason, giving advantage to GBM cells. Therefore, to better understand the role of PSA in GBM cells, we decided to lower its levels using an analog of sialic acids, namely F-NANA. This fluorinated cell-permeable sialic acid mimetic, also known as Ac_5_3F_ax_Neu5Ac, and already demonstrated to have potential effects in various cancers either in vitro or in vivo [23,24,25], was never tested in human GBM cells. Interfering with sialic acid expression in cancer could be fundamental to prevent metastasis. In murine models, desialylation of cancer cells has been achieved either by overexpressing human sialidases or by treatment with bacterial sialidases [20,45,46]. Nevertheless, gene therapy is expensive and time-consuming and sialidases from bacteria are hard to obtain with clinical grade quality. In light of these considerations, novel synthetic small inhibitors of sialic acid in cancer cells could represent a valuable therapeutic tool. Here, we show the effects of F-NANA in potently inhibiting PSA expression in U87-MG cells and confirmed these results in a self-established patient-derived GBM primary culture, GL18-15. Moreover, we report its extraordinary potential to also reduce the hypoxia-induced PSA levels in these cultures and, consequently, inhibit their migration. The action of F-NANA seems to also be specific to the inhibition of the elongation of long chains of α-2,8-linked N-acetylneuraminic acid (Neu5Ac) on NCAM protein, as we report no changes in the expression of SNA and MALII lectins both in vitro (Appendix A) and in vivo (Figure 8C,D). In light of these results, the use of this molecule may represent an important tool against an aggressive tumor such as GBM, being extremely resistant and able to rapidly spread in the surrounding parenchyma. The potential application of F-NANA in vivo, was supported by the evidence that its intranasal administration was able to reduce PSA in mouse brains. Although obtained in WT animals, these findings represent a proof of concept that such a strategy is feasible. Further studies in GBM mouse models (e.g., xenograft models) will be necessary to establish any therapeutic potential for PSA inhibition in vivo.

Unfortunately, F-NANA was not able to repress cell proliferation and to overcome hypoxia-induced resistance to TMZ. However, the strategy of lowering PSA levels in concomitance with chemotherapy cannot be excluded and could be pursued with the sole intent of preventing or simply slowing the escape of GBM cells.

Reduced degree of polysialylation, by F-NANA treatment, may interfere with NCAM intracellular signaling, which involves the activation of ERK, AKT and the wnt/beta-catenin pathways [45,46,47]. Such pathways are usually induced by the hypoxic microenvironment and are involved in the migration of cancer cells. The gradient of these factors within the tumor microenvironment, as well as the interaction of PSA-NCAM with integrins [48], may conceivably drive cancer cells towards target sites to colonize.

A solid tumor is also characterized by the so called stemness niche, a place where cancer cells experience low nutrients, hypoxia, reactive oxygen species (ROS), and low pH. All these factors contribute and favor the quiescent state in which cancer stem cells rely on to be less vulnerable, thus permitting them to survive in such extreme conditions. Trying to understand the molecular mechanisms of GBM cell differentiation is urgent, as is finding any possible strategy to interfere with their undifferentiated state. Considering the importance of this topic, therapies based on cellular differentiation have been intensively explored in recent years [10,11,49,50]. In this study, we demonstrated the involvement of PSA in the differentiation of GBM cells. In particular, we focused on the involvement of the polysialylation of NCAM either on the hypoxia-induced dedifferentiation of GBM cells or the differentiation of GSCs, thus noticing its crucial role. We report that by blocking PSA synthesis we could prevent the influence that hypoxia has in pushing GBM cells to dedifferentiate. In a similar way, when we investigated GSCs differentiation concomitant to the lowering of PSA levels, we assisted the acceleration of this event. These data may suggest that the elevation of the degree of polysialylation is necessary for those cells that need to escape the cell cycle and enter a quiescent state. To note, the dedifferentiation process induced by low oxygen concentrations requires the induction of the stemness genes Oct-4, Sox-2, Nanog and the stemness marker nestin. As concerns the expression of differentiation markers, we report the lowering of βIII-tubulin, highlighting a process involving the neuronal lineage (see also Appendix A). On the contrary, when GBM stem-like cells are forced to differentiate, this event occurs towards the glial lineage, with increasing levels of GFAP (see also Appendix A). Both mechanisms seem to be altered when PSA levels are modulated, and this may have a repercussion on the progression of the disease. PSA-NCAM has already been described as a prognostic factor in GBM, being able to regulate Olig2 expression [26]. Other studies have reported that βIII-tubulin is induced by low oxygen concentrations in GBM cells and that this event is dependent on HIF-2α [51]. In contrast, our data suggest that hypoxia can promote neuronal dedifferentiation in GBM cells and that we can partially reverse this event by inhibiting PSA levels. This result is in line with previous studies highlighting the way in which neuronal differentiation abrogates GBM aggressiveness [52]. However, we would like to underscore how, in GBM cells, it is not an anomaly to find the co-expression of glial, neuronal, and mesenchymal markers. As has already been suggested by others [49,53], our results confirm that GBM cells can move back and forth along the line of differentiation by different mechanisms depending on the location, stimuli, and microenvironment. The composition in growth factors and nutrients, as well as the pH and oxygen levels, all absolutely contribute to the constitution of an ideal niche for cancer stem cells which are characterized by aberrant glycosylation and polysialylation. In this regards, PSA-NCAM has been observed on the surface of cancer stem-like cells exposed to extreme microenvironments [50]. Interfering with PolySTs could disrupt the NCAM signaling that sustains the stemness of GBM cells.

Lowering PSA levels may have other beneficial effects. PSA has been demonstrated to have a role in the regulation of microglia activation [54]. The crosstalk between GBM and microglia cells is considered crucial for GBM growth and progression and is influenced by tumor microenvironment [55]. In the early phases of the disease, tumor infiltrating microglia cells promote tumorigenesis rather than contrast it. The margins of the tumor are also surrounded by microglial cells which have different roles depending on their polarization. M1 microglia cells have an oncosuppressive role, while M2 phenotype cells favor tumor growth and infiltration/migration [56]. We believe that it would be of fundamental importance to understand how the tumor microenvironment and the gradient of oxygen concentration from the core to the edges of the tumor can influence microglia polarization and how we can interfere in the cross-talk between GBM and microglia cells by modulating PSA.

In conclusion, we have demonstrated a pivotal role for PSA in GBM cells under hypoxia, which is outlined in Figure 9. In particular, we have shown that the hypoxic microenvironment stimulates a greater polysialylation of NCAM, which is associated with a more undifferentiated state of GBM cells as well as to a greater motility. We were also able to interfere with these processes by recurring to F-NANA, an analog of sialic acids capable of lowering the expression of PSA, which has already been demonstrated to be effective in vitro and in vivo for other tumors. In light of our results, targeting the synthesis of PSA, as well as the cells that rely on its expression, could represent a promising strategy to help limit the aggressive potential of a tumor as GBM.

## 4. Materials and Methods

### 4.1. Cell Cultures and Reagents

U87-MG, U251 and A172 human glioblastoma multiforme cell lines were purchased from CLS (Cell Lines Service GmbH, Eppelheim, Germany). Mouse glioma GL261 cell line was purchased from DSMZ (German Collection of Microorganisms and Cell Cultures GmbH, Braunschweig, Germany) and human GBM U373 cell line was purchased from ATCC (American Type Culture Collection, Manassas, VA, USA). All GBM cell lines were grown in Dulbecco’s Modified Eagle Medium (DMEM) supplemented with 10% heat-inactivated Fetal Bovine Serum (FBS, Sigma-Aldrich, St. Louis, MO, USA), 100 IU/mL penicillin G, 100 µg/mL streptomycin, 1% L-glutamine, 1% nonessential amino acids, and 1 mM sodium pyruvate at 37 °C in 5% CO_2_-humidified atmosphere. The cells were subcultured only when confluent and the medium was replaced twice a week. U87 neurospheres (U87NS) were derived as already reported by our group [57]. Briefly, U87-MG cells were grown in serum-free DMEM-F12 medium supplemented with 20 ng/mL bFGF, 20 ng/mL EGF (PeproTech, Rocky Hill, NJ, USA) B-27, and N2 supplements (Life Technologies, Carlsbad, CA, USA). U87NS were cultured for at least 10 days and the enrichment in CD133^+^ cells was evaluated by flow cytometry before proceeding for each experiment. GBM primary cultures were maintained in DMEM/F12 medium supplemented as the DMEM medium above mentioned. Hypoxia experiments were performed by incubating cells in a GasPak system (BD Biosciences, San Jose, CA, USA) and flushed with a gas mixture composed of 95% N2 and 5% CO_2_ at 37 °C. The sialyltransferase inhibitor F-NANA (3Fax-Peracetyl Neu5Ac, 566224, Sigma-Aldrich) was dissolved in Dimethyl Sulfoxide (DMSO, Sigma-Aldrich) at a final concentration of 100 mM.

### 4.2. Primary Cultures Establishment

#### 4.2.1. GBM Primary Cultures

GBM primary cultures were derived from tumor specimens of patients who were diagnosed with glioblastoma (WHO grade IV) and underwent surgical resection at Sant’Andrea Hospital of Rome (Neurosurgical Unit, “Sapienza” University of Rome). Before surgery, a signed written consent was obtained from the patient. The study was approved by the Ethical Commission of “Sapienza” University of Rome. The procedure of culture establishment has already been described by our group [58], with some modifications. Briefly, the intraoperative tissue samples were placed in DMEM/F12 medium supplemented with 5% FBS at 4 °C. Then, GBM tissues were minced with scalpel blades and mechanically dissociated with sterile glass pipettes to obtain a single-cell suspension. Before plating, the cell suspension was filtered twice with a 100 µm and again with a 70 µm cell strainer and centrifuged at 200× *g* for 5 min at room temperature. Culture medium was half changed every three/four days and the cells were sub-cultured only when total confluency was reached.

#### 4.2.2. Neonatal Rat Astrocytes

Primary rat astrocytes were established from 1 to 3-days old Wistar rats (Charles River Laboratories, L’Arbresle, France). Brains were isolated, cortices dissected, freed of the meninges, cubed into small pieces, and enzymatically digested with trypsin at 37 °C for 30 min in a water bath. After this time, brain tissues were mechanically dissociated into a single-cell suspension by pipetting 10 times with a sterile glass pipette. Then, DMEM medium supplemented with 20% FBS, 100 IU/mL penicillin G, 100 µg/mL streptomycin, 1% L-glutamine, 1% nonessential amino acids, and 1 mM sodium pyruvate was added to the cell suspension and then filtered through a 70 μm pore size cell strainer. Before seeding, cells were centrifuged at 200× *g* for 5 min and resuspended in DMEM medium supplemented with 20% FBS at a concentration of 5 × 10^5^ cells/mL. After 3 days, flasks with attached cells were vigorously agitated at 300 rpm in a rotating incubator at 37 °C for 16 h to detach microglia. After this time, the culture medium for the purified astrocytes was replaced and slowly diminished for its FBS concentration until 10%. After 2 weeks, confluent cultures were detached with trypsin, diluted, reseeded for subsequent experiments, and analyzed for intracellular GFAP staining by cytofluorimetric analysis. Only cultures above 95% of GFAP positivity received the approval for further applications.

### 4.3. MTS Assay

#### 4.3.1. Proliferation Evaluation

U87-MG cells (2 × 10^3^ cells/well) were seeded into 96-well plates and maintained overnight. Then, cells were exposed or not for 72 h to 100 µM F-NANA, either under normoxia or hypoxia. Proliferation was evaluated by MTS-formazan reduction (Promega, Madison, WI, USA) by evaluating the absorbance at 492 nm. Three independent experiments were performed in quintuplicate, and results were expressed as mean ± SD.

#### 4.3.2. Chemoresistance Evaluation

U87-MG cells (2 × 10^3^ cells/well) were seeded into 96-well plates and maintained overnight. Then, cells were exposed for 72 h to 200 µM TMZ, in presence or not of 100 µM F-NANA either under normoxia or hypoxia. Viability was measured by MTS-formazan reduction (Promega, Madison, WI, USA) by absorbance at 492 nm and indicated as percentage versus control. Three independent experiments were performed in quintuplicate, and results were expressed as mean ± SD.

### 4.4. Cytofluorimetric Analysis

Cytofluorimetric analysis was performed either by CD56-APC eFluor 780 direct staining or PSA/AlexaFluor 488 indirect staining on U87-MG cells. In brief, U87-MG cells (2 × 10^5^) were seeded into 60 mm diameter plates and maintained overnight. Then, cells were incubated or not in a hypoxic atmosphere for 72 h. Then, cells were collected, washed, and resuspended in PBS, 2% FBS (106 cells/100 µL). Samples were incubated for 30 min at 4 °C with 10 µL of mouse monoclonal anti-CD56-APC eFluor 780 (47-0567-41, eBioscience, dilution 1:10) or rabbit monoclonal anti-PSA antibody (MBS488177, MyBioSource, San Diego, CA, USA—dilution 1:10), washed and resuspended again in 100 µL of PBS, 2% FBS. The samples stained for PSA were incubated for an additional 30 min at 4 °C with AlexaFluor 488-conjugated goat anti-rabbit antibody (dilution 1:200, A11034, Life Technologies. After being additionally washed with ice cold PBS, cells were resuspended in PBS, 2% FBS, and the samples were acquired on a FACs ARIA II instrument using FACs DiVa software (v.6.1.1, both by Becton Dickinson, Milan, Italy). At least 20,000 events were recorded and analyzed using Flowing software (v2.5.1, Turku Centre for Biotechnology, University of Turku, Turku, Finland). Each experiment was performed independently three times.

### 4.5. Western Blot

Western blot analysis of U87-MG and GL18-15 cells total protein extracts was performed by lysing cell pellets in RIPA buffer (50 mM Tris–HCl pH 8.0, 150 mM NaCl, 1% Nonidet P-40, 1 mM EDTA, 0.5% sodium deoxycholate, 0.1% SDS) with protease inhibitors, 1 mM PMSF, 1 mM DTT, and 0.5 mM sodium orthovanadate (Sigma–Aldrich). Protein concentration was determined by the Bradford assay (Bio-Rad, Hercules, CA, USA). A total of 40 µg of proteins per sample were resolved on SDS–PAGE gels and blotted onto a PVDF membrane (Amersham HyBond-P GE Healthcare, Chicago, IL, USA). After blocking at room temperature in 5% dry-milk in Phosphate Buffer Saline (PBS, Sigma-Aldrich) containing 0.1% Tween-20 (Sigma-Aldrich) for 1 h, membranes were incubated overnight at 4 °C with the following primary antibodies: mouse monoclonal anti-HIF-1α (NB100-105, Novus Biologicals, Littleton, CO, USA—dilution 1:1000), rabbit polyclonal anti-NCAM (AB-90239, Immunological Sciences, Roma, Italy—dilution 1:1000), rabbit monoclonal anti-PSA (MBS488177, MyBioSource—dilution 1:1000), mouse monoclonal anti-GFAP (sc-16648, Santa Cruz Biotechnology—dilution 1:1000), mouse monoclonal anti-Nestin (sc-23927, Santa Cruz Biotechnology, Dallas, TX, USA—dilution 1:500), mouse monoclonal anti-βIII-tubulin (T5076, Sigma-Aldrich—dilution 1:2500) and mouse monoclonal anti-α-tubulin (T5168, Sigma-Aldrich—dilution 1:5000) antibody was used to normalize results. Membranes were then incubated with anti-mouse and anti-rabbit horseradish (HRP) peroxidase conjugated secondary antibodies (170-6516, 170-6515, Bio-Rad-dilution 1:10,000). Signals were detected by Clarity ECL Western Blotting substrates (170-5060, Bio-Rad). Digital images were acquired using a ChemiDoc XRS C System (BioRad). Band intensities were quantified by densitometric analysis using Image Lab software (BioRad), and the relative adjusted volumes were normalized to those of α-tubulin. Each experiment was performed independently three times and results were expressed as mean ± SD.

### 4.6. RNA Extraction and Real-Time PCR

Total RNA was isolated from U87-MG, U251 and GBM primary cultures using innuPREP RNA Mini Kit (AnalytiK Jena, Jena, Germany) according to the manufacturer’s instructions. To extract RNA, cultured cells were grown in 60 mm-diameter Petri dish and lysed when confluent. mRNA concentration was quantified using Nanodrop spectrophotometer (ThermoFisher Scientific, Waltham, MA, USA). One microgram of mRNA was converted to cDNA using the High-Capacity cDNA Reverse Transcription Kit (Applied Biosystem, Warrington, UK) according to the manufacturer’s instructions. Gene expression was quantified by real-time PCR using the 7900HT Fast Real-Time PCR System and Power SYBR Green PCR Master Mix (Applied Biosystem) according to the manufacturer’s instructions. Each experiment was independently repeated three times in triplicate and results were expressed as mean ± SD. Gene expression levels were calculated from real-time PCR data by the comparative threshold cycle (CT) method using as an internal reference the HPRT1 housekeeping gene. The following gene-specific primers were used: human ST8SiaII (STX): FW 5′-CCTCATCTTCGCAGACATCTCA-3′-RV 5′-ATCTGATTGTACCTCTGCCTCC-3′; human ST8Sia IV (PST): FW: 5′-ACTGAAAGTGCGAACTGCCT-3′-RV 5′-GAGAAGACCTGTGCTGGGTC-3′; human GFAP: FW 5′-GGCCACTGTGAGGCAGAA-3′-RV 5′-GTGGCTTCATCTGCTTCCTG-3′; human OCT4: FW 5′-GTGGAGAGCAACTCCGATG-3′-RV 5′-TCTGCAGAGCTTTGATGTCC-3′; human Sox-2: FW 5′-ATGGGTTCGGTGGTCAAGT-3′-RV 5′-GGAGGAAGAGGTAACCACAGG-3′; human Nanog: FW 5′-ATGCCTCACACGGAGACTGT-3′-RV 5′-AGGGCTGTCCTGAATAAGCA-3′; human HPRT1: FW 5′-TGATAGATCCATTCCTATGACTGTAGA-3′-RV 5′-CAAGACATTCTTTCCAGTTAAAGTTG-3′.

### 4.7. Immunohistochemical Analysis

Immunohistochemical analysis was conducted as previously described by our group [59], with some modifications. In brief, paraffin-embedded tissues of a patient diagnosed with GBM (grade IV, WHO) and of a control donor were deparaffinized in xylene, rehydrated in descending graded alcohols, incubated for 15 min in 3% H_2_O_2_ in methanol to block endogenous peroxidases activity, and then subjected for 30 min to microwave heat-induced antigen retrieval in sodium citrate buffer (10 mM tri-sodium citrate dihydrate, 0.05% Tween 20, pH 6.0). After a blocking step with Super Block reagent (ScyTek Laboratories, Logan, UT, USA) for 10 min, different serial sections were incubated overnight with mouse monoclonal anti-HIF-1α (NB100-105, Novus Biologicals—dilution 1:100), rabbit polyclonal anti-NCAM (AB-90239, Immunological Sciences—dilution 1:100) and rabbit monoclonal anti-PSA (MBS488177, MyBioSource—dilution 1:100) at 4 °C, washed three times with PBS, incubated for 10 min with UltraTek Anti-Polyvalent (ScyTek Laboratories) at room temperature, washed again three times with PBS and incubated 10 min at room temperature with UltraTek HRP (ScyTek Laboratories). Then, slides were washed three times in PBS prior to be stained with 3-3-diaminobenzidine chromogen (DAB, ScyTek Laboratories) to visualize the reaction product. Finally, slides were counterstained with hematoxylin to visualize nuclei. A Nikon Eclipse Ni motorized microscope system at 20× magnification was used to acquire images. This study was carried out according to the principles of the Helsinki Declaration and the protocols approved by the ethics committee.

### 4.8. Immunofluorescence Analysis

U87-MG cells (5 × 10^3^) were plated on 8-well chamber slides (Nunc Lab-Tek, Waltham, MA, USA) and maintained overnight in DMEM supplemented with 10% FBS either under hypoxia or normoxia, in presence or not of 100 μM F-NANA. For immunofluorescence staining we followed the protocol previously described by our group [60]. Briefly, cells were fixed by 4% paraformaldehyde, permeabilized with 0.3% PBS-triton X-100, blocked for 1 h with 0.2% PBS-gelatin and incubated overnight at 4 °C with the following primary antibodies: rabbit monoclonal anti-PSA (MBS488177, MyBioSource—dilution 1:200) and mouse monoclonal anti-βIII-tubulin (T5076, Sigma-Aldrich—dilution 1:300). Goat anti-rabbit Alexa-fluor 488 and goat anti-mouse Alexa-fluor 594 secondary antibodies (dilution 1:1000, Life Technologies) were used and nuclei were counterstained by incubating cells with DAPI for 3 min in the dark. A Nikon Eclipse Ni motorized microscope was used to acquire the images at 20× magnification.

### 4.9. Differentiation Assay

U87NS at day 14 were washed, centrifuged at 200× *g*, resuspended in DMEM supplemented with 10% FBS in presence or absence of 100 µM F-NANA and allowed to differentiate for 5 days in 60 mm diameter tissue culture petri dishes. Total RNA was extracted, and RT-PCR was performed to evaluate mRNA expression of the stemness markers oct-4, sox-2, and nanog. For immunofluorescence analysis, the assay was performed in 8-well chamber slides (Nunc Lab-Tek) and at the end of the experiment cells were fixed with 4% paraformaldehyde, permeabilized with 0.3% PBS-triton X-100, blocked for 1 h with 0.2% PBS-gelatin and incubated overnight at 4 °C with the following primary antibodies: rabbit monoclonal anti-PSA (MBS488177, MyBioSource—dilution 1:200), mouse monoclonal anti-GFAP (sc-16648, Santa Cruz Biotechnology—dilution 1:500), mouse monoclonal anti-Nestin (sc-23927, Santa Cruz Biotechnology—dilution 1:200), mouse monoclonal anti-βIII-tubulin (T5076, Sigma-Aldrich—dilution 1:200). The secondary antibodies used were goat anti-mouse Alexa-fluor 594 conjugate and goat anti-rabbit Alexa-fluor 488 conjugate (dilution 1:1000, Life Technologies). Nuclei were counterstained by DAPI incorporation for 3 min in the dark and Mowiol was used to mount coverslips on chamber slides. Images were acquired with a Nikon Eclipse Ni motorized microscope system at 20× magnification.

### 4.10. Animal Models

All experimental procedures were approved by the IRCCS Neuromed Animal Care Review Board and by “Istituto Superiore di Sanità” (ISS permit numbers: 760/2020-PR) and were conducted according to 2010/63/EU Directive for animal experiments. Female wild-type mice (strain name: B6CBAF1/J) were used for all in vivo experiments.

#### 4.10.1. In Vivo Drug Administration

F-NANA (Methyl 5-acetamido-2,4,7,8,9-penta-O-acetyl-3,5-dideoxy-3-fluoro-D-erythro-β-L-manno-2-nonulopyranosate) [21,22] was dissolved in DMSO, further diluted in saline (vehicle) and daily administered by intranasal (i.n.) injection [61] at concentration of 100 µM. Control mice were injected daily with the same volume of vehicle-containing DMSO.

#### 4.10.2. Motor Behavior Tests

Motor performance was assessed by Rotarod tests as previously described [62]. All animals used for biochemical experiments were euthanized after 3 weeks of treatment (7 weeks of age).

#### 4.10.3. Brain Lysate Preparation Immunoblottings

Mice were sacrificed by cervical dislocation and brains were immediately snap-frozen in liquid N_2_ and pulverized in a mortar with a pestle as previously described [62].

Proteins (20 μg) were resolved on 8% SDS-PAGE and immunoblotted with the following antibodies: anti-PSA (MBS488177, MyBioSource—dilution 1:1000) and anti-NCAM (AB-90239, Immunological Sciences—dilution 1:1000). For protein normalization, anti-β-actin (A5441, Sigma Aldrich—dilution 1:2000) was used. Immunoblots were then exposed to specific HRP-conjugated secondary antibodies. Protein bands were visualized by ECL and quantified by Image Lab Software (Bio-Rad Laboratories).

For the analysis of SNA and MAL II, protein lysates (20 μg) were resolved on 8% SDS-PAGE and the membrane was blocked with a Carbo Free Blocking Solution (1X) (SP-5040, Vector Laboratories, Burlingame, CA, USA). Membrane was immunoblotted with anti-SNA antibody (B-1305, Vector Laboratories—dilution 1:1000) and anti-MAL II antibody (B-1265, Vector Laboratories—dilution 1:1000) in a Carbo Free Blocking Solution (1X). A Streptavidin conjugated with Peroxidase antibody (IS-6633-1, Immunological Sciences) was used as secondary antibody. Protein bands were visualized by ECL and quantified as described above.

### 4.11. Statistical Analysis

All statistical analyses were performed using GraphPad Prism v.7 software. Results are expressed as percentage of the mean ± standard deviation (SD). In all cases, data were analyzed by one-way analysis of variance (ANOVA). A *p*-value < 0.05 was considered as statistically significant.

## Figures and Tables

**Figure 1 ijms-23-09563-f001:**
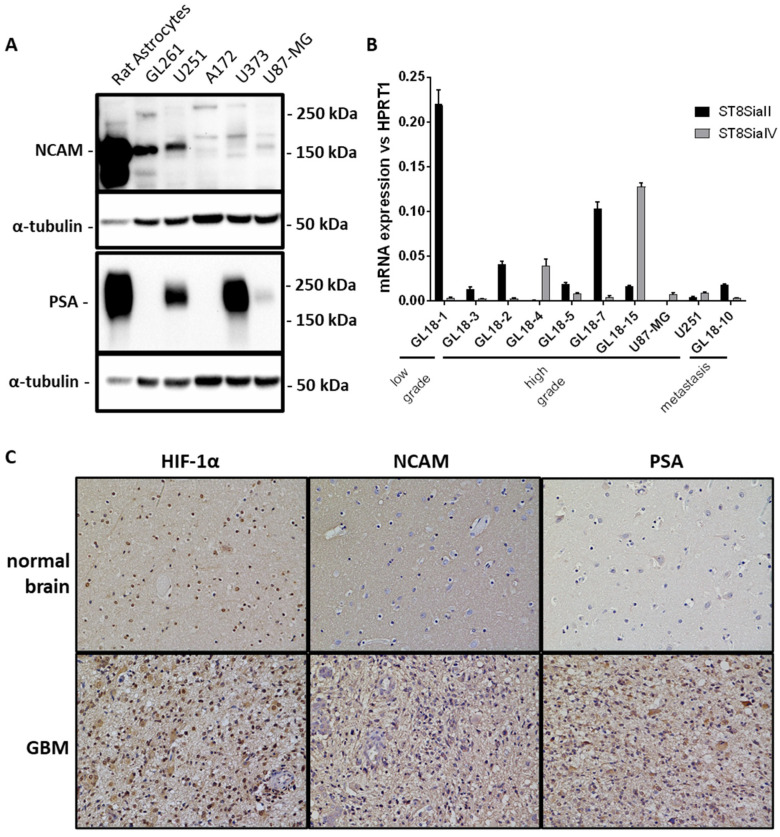
NCAM and PSA expression in different GBM cell cultures and hypoxic areas of GBM tissues. (**A**) Western blot qualitative analysis showing NCAM isoforms and PSA expression levels in a mouse glioma cell line (GL261) and four human GBM cell lines (U251, A172, U373, U87-MG) compared to normal neonatal rat astrocytes, included as positive control. (**B**) Real-time PCR analysis showing mRNA expression of polysialyltransferases II and IV (ST8SiaII and ST8SiaIV) in low grade (GL18-1 and GL18-3) and high grade (GL18-2, GL18-4, GL18-5, GL18-7, GL18-15) glioma patient-derived primary cultures compared to U87-MG and U251 GBM cell lines and a brain metastasis. (**C**) Immunohistochemical analysis showing the expression of NCAM, PSA and HIF-1α in formalin-fixed, paraffin embedded sections of a GBM tissue compared to a normal, non-tumoral brain.

**Figure 2 ijms-23-09563-f002:**
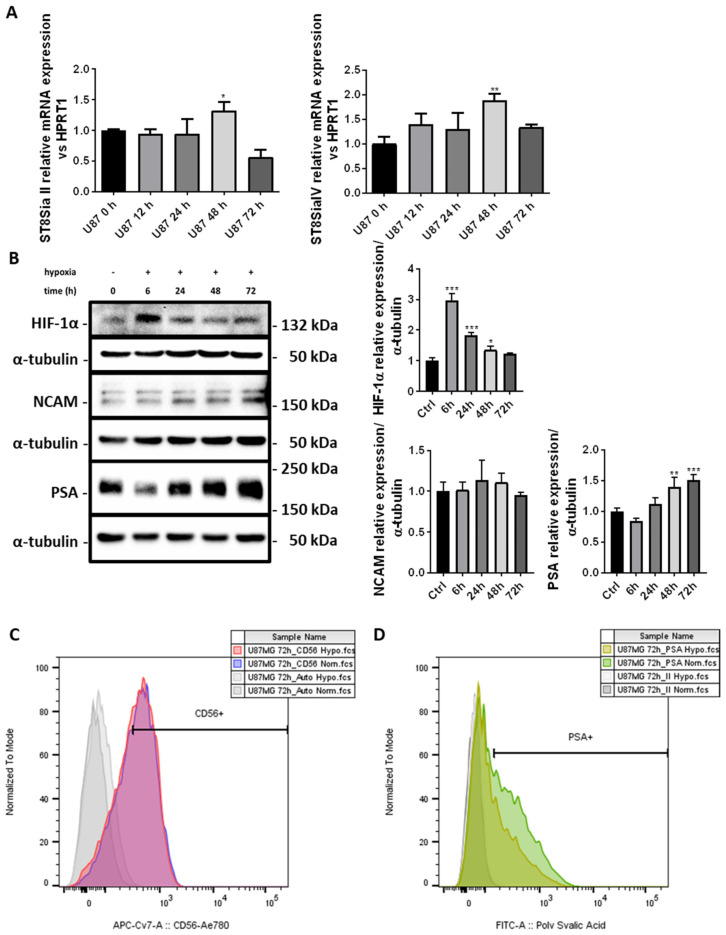
PolySTs, NCAM and PSA expression in U87-MG cells under chronic hypoxia. (**A**) Real-Time PCR analysis showing the time-course (0–72 h) of mRNA expression of polysialyltransferases II and IV (ST8SiaII and ST8SiaI(V) in U87-MG cells exposed to low oxygen concentrations. (**B**) Western blot analysis showing the time-course (0–72 h) of HIF-1α activation, NCAM and PSA expression in U87-MG cells under hypoxia. (**C**) Cytofluorimetric analysis showing the extracellular expression of NCAM (CD56) in U87-MG cells exposed for 72 h either to chronic hypoxia (red histogram) or normoxia (blue histogram). (**D**) Cytofluorimetric analysis showing the extracellular expression of PSA in U87-MG cells exposed for 72 h either to chronic hypoxia (light green histogram) or normoxia (green histogram). All results are expressed as the mean ± SD of three independent experiments. *: *p* < 0.05; **: *p* < 0.01; ***: *p* < 0.001.

**Figure 3 ijms-23-09563-f003:**
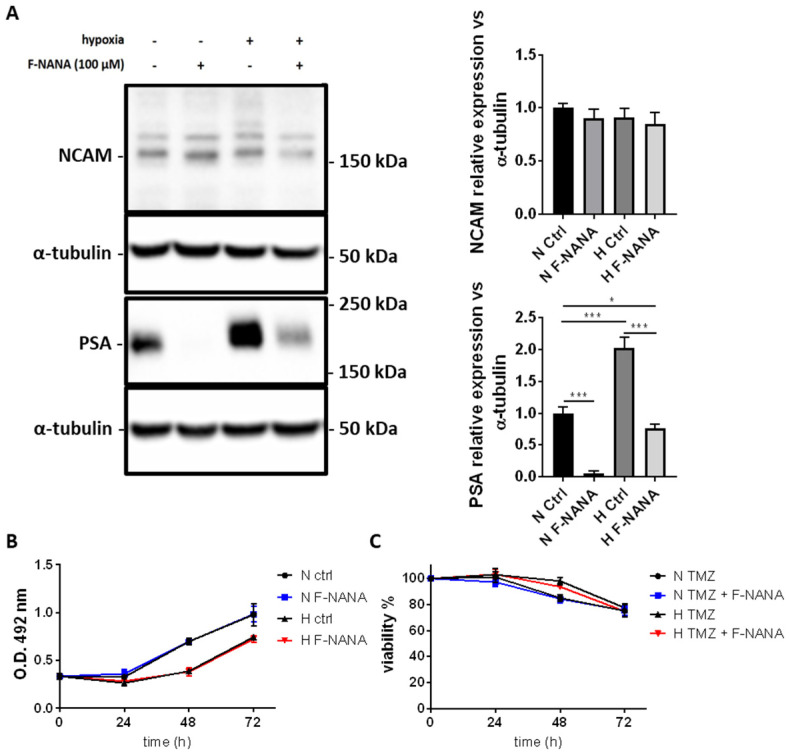
Effects of F-NANA on the proliferation, chemoresistance to TMZ and levels of NCAM and PSA in U87-MG cells under hypoxia. (**A**) Western blot analysis showing the expression of NCAM and PSA in U87-MG cells grown either under normal or low oxygen concentrations, in presence or not of 100 μM F-NANA. (**B**) MTS assay showing the proliferation (expressed as absorbance at 492 nm) of U87-MG cells exposed for 72 h either to chronic hypoxia or normoxia, in presence or not of 100 μM F-NANA. (**C**) Viability evaluation in U87-MG cells grown either in normoxia or hypoxia and treated with 200 μM TMZ, in presence or not of 100 μM F-NANA. All results are expressed as the mean ± SD of three independent experiments. *: *p* < 0.05; ***: *p* < 0.001.

**Figure 4 ijms-23-09563-f004:**
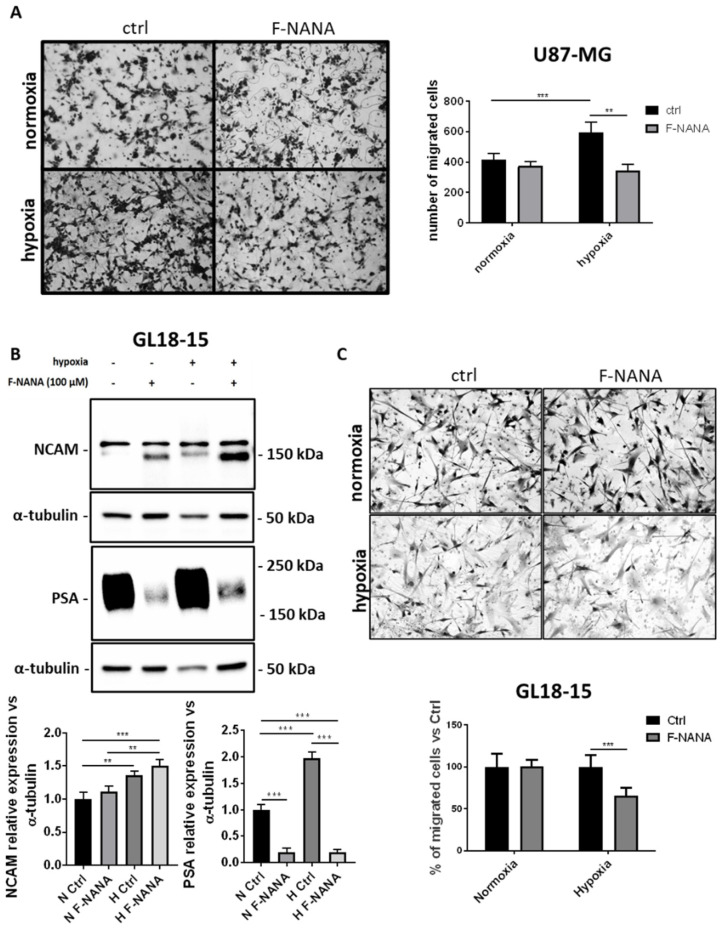
Effect of PSA inhibition on the migration of GBM cells under hypoxic conditions. (**A**) Transwell migration assay showing the number of migrated U87-MG cells under normal and low oxygen concentration, in presence or not of 100 μM F-NANA. (**B**) Western blot analysis showing the expression of NCAM and PSA in a patient-derived GBM primary culture (GL18-15) exposed to 72 h chronic hypoxia, in presence or not of 100 μM F-NANA. (**C**) Transwell migration assay showing the percentage versus control of migrated GL18-15 cells under normal and low oxygen concentration, in presence or not of 100 μM F-NANA. Results are presented as the mean ± SD of three independent experiments. **: *p* < 0.01; ***: *p* < 0.001.

**Figure 5 ijms-23-09563-f005:**
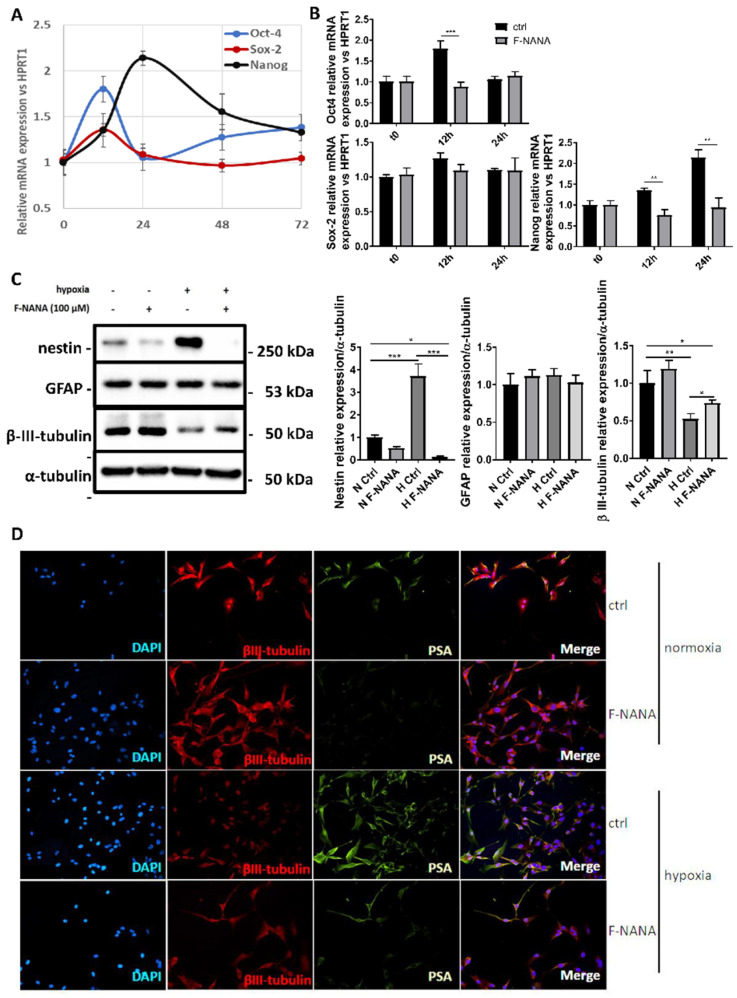
Effect of PSA inhibition on the hypoxia-induced de-differentiation of U87-MG cells. (**A**) Real-time PCR analysis showing the mRNA induction of the stemness genes Oct-4, Sox-2 and Nanog in U87-MG cells grown under hypoxic conditions for 72 h. (**B**) Real-time PCR analysis showing the effect of 100 μM F-NANA on the expression of Oct-4, Sox-2 and Nanog mRNAs after 12 and 24 h of chronic hypoxia. (**C**) Western blot analysis of the stemness marker Nestin and the differentiation markers GFAP and βIII-tubulin in U87-MG cells grown either under normoxia or hypoxia, in presence or not of 100 μM F-NANA. (**D**) Immunofluorescence analysis showing the expression of βIII-tubulin (in red) and PSA (in green) of U87-MG cells grown either under hypoxic or normoxic conditions for 72 h, in presence or not of 100 μM F-NANA. Nuclei are stained with DAPI (in blue), and merged channels are shown (Merge). Results are presented as the mean ± SD of three independent experiments. *: *p* < 0.05; **: *p* < 0.01; ***: *p* < 0.001.

**Figure 6 ijms-23-09563-f006:**
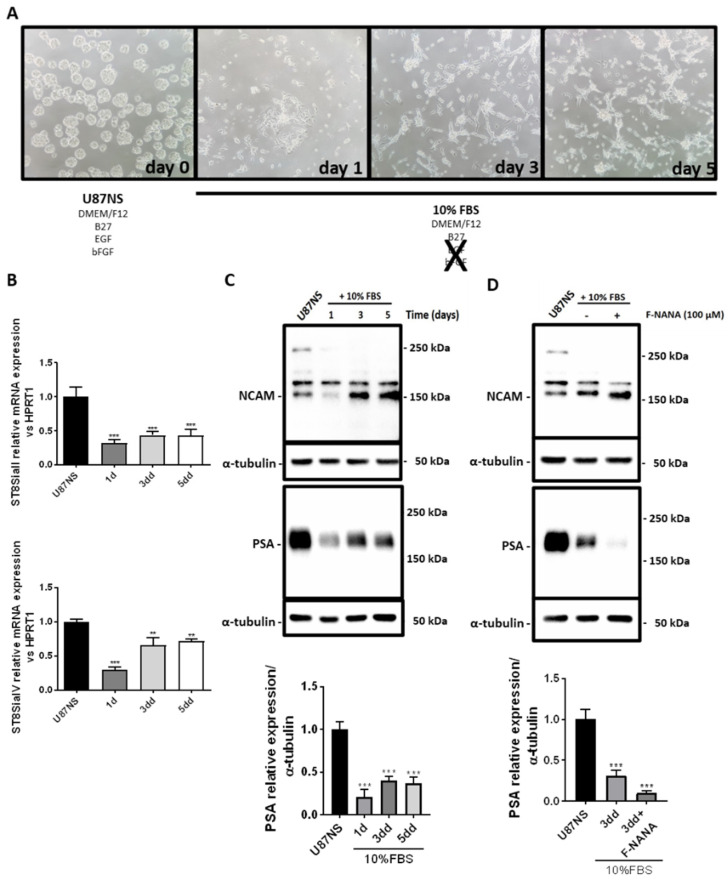
PolySTs, NCAM and PSA expression during U87NS cell differentiation. (**A**) Phase-contrast micrographs showing floating U87NS differentiated in medium supplemented with 10% FBS for 5 days. Images were acquired at day 0, 1, 3 and 5. (**B**) Real-time PCR analysis of ST8SiaII and IV mRNAs expression in U87NS differentiated for 5 days in medium supplemented with 10% FBS. (**C**) Western blot analysis showing the levels of NCAM and PSA in U87NS differentiated for 5 days in medium supplemented with 10% FBS. (**D**) Western blot analysis of NCAM and PSA expression in U87NS differentiated for 3 days in medium supplemented with 10% FBS, in presence or not of 100 μM F-NANA. Results are presented as the mean ± SD of three independent experiments. 1d: day 1; 3dd: day 3; 5dd: day 5. **: *p* < 0.01; ***: *p* < 0.001.

**Figure 7 ijms-23-09563-f007:**
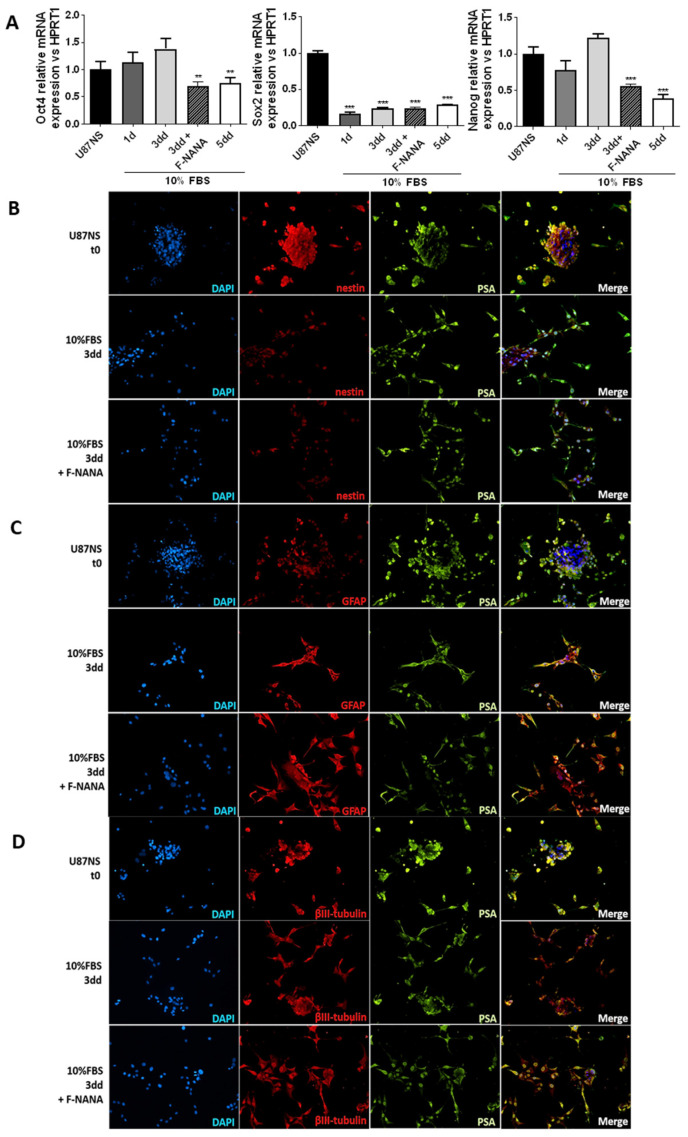
Effect of PSA inhibition on U87NS cell differentiation. (**A**) Real-time PCR analysis showing mRNA expression of the stemness genes Oct-4, Sox-2 and Nanog in U87NS cells differentiated for 5 days in medium supplemented with 10% FBS. The condition of differentiating U87NS cells treated with 100 μM F-NANA for 3 days is also reported. Results are presented as the mean ± SD of three independent experiments. **: *p* < 0.01; ***: *p* < 0.001. (**B**–**D**) Immunofluorescence analysis showing the co-expression (Merge) of PSA (in green) with the stemness marker Nestin (red) or the differentiation markers GFAP and βIII-tubulin (in red) in U87NS cells, differentiated or not for 3 days in medium supplemented with 10%FBS, treated or not with 100 μM F-NANA. Nuclei were counterstained with DAPI (in blue). 1d: day 1; 3dd: day 3; 5dd: day 5.

**Figure 8 ijms-23-09563-f008:**
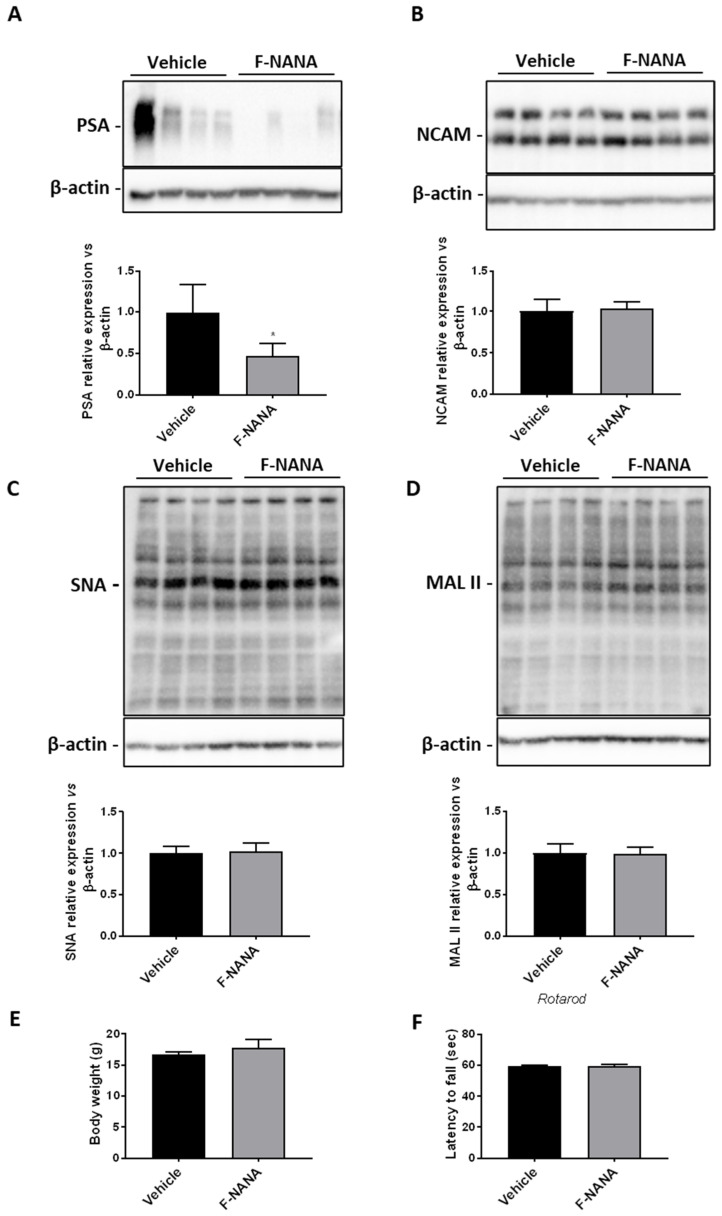
F-NANA reduces the levels of PSA in brain tissues from wild-type mice. Representative cropped Western blottings and densitometric analysis of PSA (**A**) and NCAM (**B**) levels in brain tissues from vehicle- and F-NANA-treated wild-type mice at 7 weeks of age. Data are represented as mean  ±  SD, *N*  =  5 for each group of mice. *, *p*  <  0.05; (Unpaired *t*-test). Representative cropped Western blottings and densitometric analysis of SNA (**C**) and MAL II (**D**) levels in brain tissues from the vehicle- and F-NANA-treated wild-type mice at 7 weeks of age. Data are represented as mean  ±  SD, *N*  =  5 for each group of mice. (**E**) Mouse body weight measured at the end of the treatment. Values are represented as mean ± SD. (*N* = 5 for each group of mice). (Unpaired *t*-test). (**F**) Motor performance assessed by Rotarod test. Values are represented as mean ± SD. (*N* = 5 for each group of mice). (Unpaired *t*-test).

**Figure 9 ijms-23-09563-f009:**
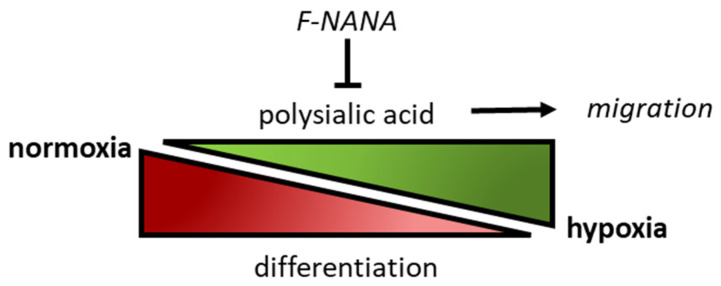
Schematic representation of the role of PSA and the effects of its inhibition in GBM cells. The figure shows the inverse correlation between the levels of PSA (polysialic acid) and the degree of differentiation of GBM cells. Low oxygen concentrations upregulate the levels of PSA in GBM cells, thus inducing their dedifferentiation and migration. All these events can be inhibited by interfering with the synthesis of PSA by treating GBM cells with F-NANA.

## Data Availability

Main data generated or analyzed in this study are included in this article. Details are available from the corresponding author on reasonable request.

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
