# Peer review of "Polysialic Acid Sustains the Hypoxia-Induced Migration and Undifferentiated State of Human Glioblastoma Cells"

_ijms, 2022, doi:10.3390/ijms23179563_

Round 1
Reviewer 1 Report
The authors investigated the role of the inhibition of polysialic acid (PSA), a carbohydrate primarily attached to the Neural Cell Adhesion Molecule (NCAM), on GBM cells exposed to hypoxia. The found that, by lowering PSA levels with the sialic acid analog, F-NANA, GBM cell migration was inhibited and the differentiation influenced by the hypoxic microenvironment. The authors concluded that PSA could represent a novel potential molecular target for GBM therapy.
The topic is interesting. The methods and results are also well explained and described.
I have some concerns:
1. The authors should discuss about the role of tumor microenvironment and microglia-glioma cross-talk. It is actually considered one of the main mechanisms through which GBM grows and progresses. Please see:
PMID 33561993
PMID: 35654889
2. Please update the terminology according to 5th Edition 2021 of WHO classification of CNS tumors (i.e. arabic numbers, IDH status)
Author Response
We thank reviewer 1 for his/her precious suggestions.
The authors investigated the role of the inhibition of polysialic acid (PSA), a carbohydrate primarily attached to the Neural Cell Adhesion Molecule (NCAM), on GBM cells exposed to hypoxia. The found that, by lowering PSA levels with the sialic acid analog, F-NANA, GBM cell migration was inhibited and the differentiation influenced by the hypoxic microenvironment. The authors concluded that PSA could represent a novel potential molecular target for GBM therapy.
The topic is interesting. The methods and results are also well explained and described.
I have some concerns:
- The authors should discuss about the role of tumor microenvironment and microglia-glioma cross-talk. It is actually considered one of the main mechanisms through which GBM grows and progresses. Please see:
PMID 33561993
PMID: 35654889
- We have added this fundamental mechanism about microglia-GBM cross-talk and discussed about its importance and PSA role in the discussion section. We also added references 54, 55, 56.
- Please update the terminology according to 5th Edition 2021 of WHO classification of CNS tumors (i.e. arabic numbers, IDH status)
- We feel terribly sorry to have used an old terminology. According to the 5th Edition of the WHO classification of CNS tumors (2021) we have promptly updated it in the abstract and in the introduction section and added the reference number 2.
We hope to have fully addressed all the points the referee 1 has asked us and we are sure the manuscript will be improved after this revision process.
Reviewer 2 Report
Dear All,
The authors investigated the effects of hypoxia in glioblastoma tumors. Their work discovered that hypoxia triggers polysialylation on NCAM in U87-MG and primary glioblastoma cells grown in culture. F-NANA was used to remove or inhibit polysialylation and demonstrated that this inhibition/removal causes a loss of dedifferentiation and cell migration in a hypoxic microenvironment. In addition, polysialylation was shown to be required for glioblastoma stem-like cells to maintain their de-differentiated state. Furthermore, that polysialylation can be controlled in vivo.
The manuscript should be acceptable with minor revisions.
1) The authors should explain how the loss of polysialylation by F-NANA causes inhibition of hypoxia-induced migration.
2) The authors also need to elaborate on how polysialylation of NCAM governs the so-called stemness (dedifferentiation state) of glioblastoma stem-like cells.
For both 1 and 2, a discussion needs to be provided of putative signal transduction pathways involved in controlling migration and differentiation that elucidate the significant mechanisms governing cell state and mobility.
3) Minor point: Lines 249-259 do not mention Figures 7C and 7D.
Author Response
Dear All,
The authors investigated the effects of hypoxia in glioblastoma tumors. Their work discovered that hypoxia triggers polysialylation on NCAM in U87-MG and primary glioblastoma cells grown in culture. F-NANA was used to remove or inhibit polysialylation and demonstrated that this inhibition/removal causes a loss of dedifferentiation and cell migration in a hypoxic microenvironment. In addition, polysialylation was shown to be required for glioblastoma stem-like cells to maintain their de-differentiated state. Furthermore, that polysialylation can be controlled in vivo.
The manuscript should be acceptable with minor revisions.
- The authors should explain how the loss of polysialylation by F-NANA causes inhibition of hypoxia-induced migration.
- The authors also need to elabo on how polysialylation of NCAM governs the so-called stemness (dedifferentiation state) of glioblastoma stem-like cells.
For both 1 and 2, a discussion needs to be provided of putative signal transduction pathways involved in controlling migration and differentiation that elucidate the significant mechanisms governing cell state and mobility.
Reply. We thank the Reviewer for his/her suggestions. We addressed these points in the discussion of the revised version of the manuscript.
Minor point: Lines 249-259 do not mention Figures 7C and 7D.
Reply. We addressed this issue in the revised version of the manuscript.
Reviewer 3 Report
The present manuscript describes interesting data that leads the authors to conclude that PSA could plausibly represent a new molecular target for the development of alternative pharmacological strategies to manage a devastating tumor like GMB.
major point
the main problem I see on the manuscript is thet the authors state (line 272- 273) that "in order to provide a proof con concept that F-NANA administration might represent a potential therapeutic approach in vitro..." they carry out experiment treating 4 week-old mice with F-NANA, and the results are quite interesting, however, I believe that to make the point it is necessary, now that we know by their data that F-NANA has an effect in vivo, to inject into the brain of a set of mice, cells from a GBM, and then make to treatment groups one with F-NANA and another with vehicle, and determine at some time point if there is a difference in tumor size due to the F-NANA treatment. If there is a difference fine, otherwise, the experiment still should be added and interpreted, since as shown by the current data the F-NANA has an effect in vivo. I strongly feel that this experiment will strengthen the manuscript, independently of the result, and if there is a difference in tumor size, the authors can come stronger with their conclusions.
minor points
on line 86 where it says "in the main cell lines used in GBM research" change to "in the cell lines mainly used in GBM research"
on line 101 change "In the same day" for "Similarly," or something equivalent
In Figure 3C the scale on the Y axis should be changed from 60-100 to 0-100 since the current scale is misleading
on line 177 change "we assisted to the increase" for "concomitant with the increase..." or equivalent
on line 231-232 where it says "in-vestigated the expression of the during the differentiation of", there is something missing on the phrase, please modify
In Figure 6 (and subsequent figures) the authors start describing samples with 1d, 3dd, 5dd, please describe at least the first time the meaning, I believe that is 1 day, 3 days and 5 days after starting the differentiation process but it might be more consistent if 1day is described also as 1dd, but please explain
on the legend of figure 6 A) line 239-40, consider changing "photographs" by " Phase-contrast micrographs"
Author Response
We thank reviewer 2 for his/her time reviewing our manuscript and precious suggestions.
The present manuscript describes interesting data that leads the authors to conclude that PSA could plausibly represent a new molecular target for the development of alternative pharmacological strategies to manage a devastating tumor like GMB.
major point
the main problem I see on the manuscript is thet the authors state (line 272- 273) that "in order to provide a proof con concept that F-NANA administration might represent a potential therapeutic approach in vitro..." they carry out experiment treating 4 week-old mice with F-NANA, and the results are quite interesting, however, I believe that to make the point it is necessary, now that we know by their data that F-NANA has an effect in vivo, to inject into the brain of a set of mice, cells from a GBM, and then make to treatment groups one with F-NANA and another with vehicle, and determine at some time point if there is a difference in tumor size due to the F-NANA treatment. If there is a difference fine, otherwise, the experiment still should be added and interpreted, since as shown by the current data the F-NANA has an effect in vivo. I strongly feel that this experiment will strengthen the manuscript, independently of the result, and if there is a difference in tumor size, the authors can come stronger with their conclusions.
We kindly thank reviewer 2 for his/her suggestion. This is an important point for us and we totally agree with reviewer 2. We feel sorry that we haven’t fully satisfied the request of reviewer 2, but at the moment we cannot perform this kind of experiments that go beyond the scope of the present article. In vivo investigations have already been planned by our group and are already under consideration for a bigger project that we intend to pursue in the near future. In the present article we only anticipated and wanted to share this information (important for us) that we are able to modulate PSA in vivo (in the brain) and we will for sure keep on going with this in the next study taking into consideration also the suggested advice of the reviewer 2.
minor points
on line 86 where it says "in the main cell lines used in GBM research" change to "in the cell lines mainly used in GBM research"
- We have made the suggested correction.
on line 101 change "In the same day" for "Similarly," or something equivalent
- We have made the suggested correction.
In Figure 3C the scale on the Y axis should be changed from 60-100 to 0-100 since the current scale is misleading
- We have corrected figure 3 and replaced it.
on line 177 change "we assisted to the increase" for "concomitant with the increase..." or equivalent
- We have made the suggested correction.
on line 231-232 where it says "in-vestigated the expression of the during the differentiation of", there is something missing on the phrase, please modify
- We have made the suggested correction.
In Figure 6 (and subsequent figures) the authors start describing samples with 1d, 3dd, 5dd, please describe at least the first time the meaning, I believe that is 1 day, 3 days and 5 days after starting the differentiation process but it might be more consistent if 1day is described also as 1dd, but please explain
- We have added this explanation in the figure legends of figures 6-7.
on the legend of figure 6 A) line 239-40, consider changing "photographs" by " Phase-contrast micrographs"
- We have made the suggested correction.
We hope to have fully addressed all the points the referee 2 has asked us and we are sure the manuscript will be improved after this revision process.
Round 2
Reviewer 1 Report
The authors improved the manuscript that is now suitable for publication
Author Response
We kindly thank the Reviewer 1 for his/her time in reviewing our manuscript.
Reviewer 3 Report
Since the authors decided not to carry the experiment suggested by the referee, I do believe that the data does not fully supports the statements made by the authors and therefore I cannot accept the manuscript in its present form
Author Response
We understand the Reviewer's concern, therefore, according to the editor's suggestions we mitigated the conclusion of the manuscript.